# Fabrication of Partially Etched Polystyrene Nanoparticles

**DOI:** 10.3390/polym15071684

**Published:** 2023-03-28

**Authors:** In Hwan Jung, Jieun Lee, Seung Soo Shin, Youn-Jung Kang, Tae Seok Seo, Bum Jun Park

**Affiliations:** 1Department of Chemical Engineering (BK21 FOUR Integrated Engineering Program), Kyung Hee University, Yongin 17104, Gyeonggi-do, Republic of Korea; jih0221@khu.ac.kr (I.H.J.); jieun9913@khu.ac.kr (J.L.); tlstmdtn1229@khu.ac.kr (S.S.S.); 2Department of Biochemistry, Research Institute for Basic Medical Science, School of Medicine, CHA University, Seongnam-si 13496, Gyeonggi-do, Republic of Korea

**Keywords:** partially etched nanoparticles, emulsion polymerization, delayed crosslinking, Pickering emulsion, enhanced stability

## Abstract

Non-spherical polymer nanoparticles (NPs) have gained attention in various fields, but their fabrication remains challenging. In this study, we present a simple protocol for synthesizing partially etched polystyrene (PS) nanoparticles through emulsion polymerization and chemical etching. By adjusting the degree of crosslinking, we selectively dissolve the weakly crosslinked portions of the particles, resulting in partially etched PS NPs with increased surface area. These partially etched NPs are evaluated for their use as solid surfactants in Pickering emulsions, where they demonstrate significantly improved emulsion stability compared to intact spherical NPs. Our results contribute to the field of nanoparticle shape control and provide insights into developing novel materials for various applications, particularly in the area of solid surfactant usage. Additionally, the importance of conducting cellular toxicity studies using these partially etched NPs for future work is also emphasized.

## 1. Introduction

Polymer nanoparticles (NPs) have gained significant attention in recent years due to their diverse range of applications in electronics [1], photonics [2], sensors [3,4], and pharmaceuticals [5,6], owing to their versatility and tunable properties. Depending on their shapes, polymer NPs can be classified into spherical, non-spherical, and mesoporous forms. Among them, non-spherical polymer NPs have shown significant potential in biomedical engineering, offering advantages over their spherical counterparts. Specifically, non-spherical NPs have demonstrated improved cellular uptake [7], enhanced drug delivery [8], and more efficient imaging contrast agents [9]. These features are in part attributed to the unique surface properties of non-spherical NPs, such as their increased surface area and surface roughness, which provide more opportunities for interactions with cells and biomolecules.

Non-spherical NPs can be further classified into ellipsoidal, dumbbell, rod, disk, cube, and other non-spherical shapes [10,11]. However, producing non-spherical NPs is challenging as they tend to prefer a spherical shape during synthesis to minimize surface energy. Recent advancements in particle manufacturing technology have led to significant progress in developing various types of non-spherical NPs. For example, ellipsoidal NPs were produced by mechanically elongating spherical polystyrene (PS) precursor particles upon heating them above their glass transition temperature [12,13]. Dumbbell-shaped PS NPs were obtained by emulsion polymerization of styrene monomers in the presence of shape-directing agents [14,15]. In addition, preformed mesoporous silica NPs were used as a template for synthesizing polymer NPs of various shapes [16]. Furthermore, microfluidic devices facilitated precise control over reactant flows and the size and shape of the resulting polymer particles [17,18]. However, research on partially and/or randomly deformed NPs manufacturing techniques has been limited. These particles can maintain the characteristics of spherical particles while increasing surface area and improving interfacial adsorption capability, which can enhance the stability of Pickering emulsions [19,20]. Additionally, NPs with random shapes can be utilized as model particles mimicking microplastic particles found in nature for cellular toxicity studies that have caused serious environmental problems [21,22].

In this study, we focus on developing a simple protocol to fabricate partially etched PS NPs using emulsion polymerization and chemical etching. The degree of crosslinking was controlled by adjusting the time to add a crosslinking reagent during the polymerization reaction. Then, the obtained partially crosslinked PS NPs were chemically etched to selectively dissolve weakly crosslinked portions. These particles with increased surface area were evaluated by using them as solid surfactants to form Pickering emulsions. Note that Pickering emulsions are stabilized by solid particles adsorbed at a fluid–fluid interface, which are more stable and robust than traditional emulsions stabilized by molecular surfactants [23,24]. Thus, developing solid surfactants with tailored properties is of great interest for the stabilization of Pickering emulsions. Our approach provides insights into the potential of partially etched NPs as solid surfactants in emulsion stabilization. Furthermore, the ability to control the degree of surface roughness and surface area of the NPs offers new opportunities for developing novel materials with tailored properties for various medical and environmental applications.

## 2. Materials and Methods

### 2.1. Materials

For emulsion polymerization, ultrapure water (resistivity ≥ 18.2 MΩ·cm, Aquapuri5, Young In Scientific Co., Ltd., Seoul, Republic of Korea) was used as an aqueous solution. Styrene monomer (99.5%) was purchased from Samchun Chemicals (Seoul, Republic of Korea). Sodium dodecyl sulfate (SDS, ≥99.0%, Sigma-Aldrich, St. Louis, MO, USA) and cetyltrimethylammonium bromide (CTAB, 99+%, Acros Organics, Fair Lawn, NJ, USA) were used as surfactants. Initiator (potassium persulfate (PPS), K_2_S_2_O_8_, ≥99.0%) and crosslinker (divinyl benzene (DVB), 80%) were obtained from Sigma-Aldrich (St. Louis, MO, USA). Etching solvents used were tetrahydrofuran (THF, extra pure, Daejung Chemicals & Metals, Siheung-si, Gyeonggi-do, Republic of Korea), toluene (99.9%, Sigma-Aldrich, St. Louis, MO, USA), and chloroform (≥99.0%, Sigma-Aldrich, St. Louis, MO, USA). Ethanol (94.5%) was obtained from Daejung Chemicals & Metals (Siheung-si, Gyeonggi-do, Republic of Korea). For Pickering emulsion formation, n-decane (99+%, Acros Organics, Fair Lawn, NJ, USA) was used as an oil phase, which was filtered using aluminum oxide particles (Al_2_O_3_, Pore size = 58 Å, Sigma-Aldrich, St. Louis, MO, USA) to remove any polar impurities. All chemicals were used as received unless otherwise noted.

### 2.2. Emulsion Polymerization of PS NPs

Emulsion polymerization was used to synthesize PS NPs [25]. A 500 mL three-necked round bottom flask containing 200 mL of ultrapure water was placed in a silicon oil bath at 90 °C (Figure 1a). To prevent solvent evaporation, a reflux condenser was attached to the flask. After adding 1.04 × 10^−3^ mol of surfactant and 10 mL of styrene monomer to the reactor, N_2_ purging was performed for 30 min. The polymerization process was initiated by adding the PPS initiator. After 18 h of polymerization, PS NPs were obtained and subsequently washed with ethanol. This was achieved by using a series of centrifugation at 14,000 rpm and redispersion steps, repeated five times at 5 °C to eliminate residual reactants. The synthesis of partially crosslinked NPs followed the same protocol, with the exception that 4 wt% DVB was added to the reactor at a designated time $t_{DVB}$ after PPS addition (Figure 1b).

### 2.3. Chemical Etching of Partially Crosslinked PS NPs

An etching solvent such as THF, chloroform, or toluene was mixed with the particle solution (3 wt%) at a volume ratio of 6:1. After vortexing for 1 min, the particle solution was centrifuged at 14,000 rpm for 10 min. The resulting precipitate was redispersed in ethanol to obtain partially etched PS NPs. The same procedure was repeated for the other $t_{DVB}$ particle samples. The particle shape was observed using a scanning electron microscope (SEM, LEO SUPRA 55, Carl Zeiss, Oberkochen, Germany).

### 2.4. Fabrication of Pikckering Emulsions

The aqueous dispersion of PS NPs (9 wt%) and n-decane were mixed in a 1:1 volume ratio to prepare an oil-in-water Pickering emulsion by vortexing for 1 min. The stability of the emulsion was monitored using an inverted microscope (ECLIPSE Ti, Nikon, Tokyo, Japan) and a camera (EOS700D, Canon, Tokyo, Japan) for a period of 2 months.

## 3. Results

### 3.1. Synthesis of PS NPs

Emulsion polymerization was used to synthesize non-crosslinked and partially crosslinked PS NPs with uniform size distributions, as shown in Figure 2. For the partially crosslinked PS NPs, the crosslinking reagent DVB was added approximately at $t_{DVD}=$ 70 min, which corresponds to ~80% of the final size [26], following the initiator addition. When using the anionic surfactant SDS, the mean diameters of non-crosslinked and partially crosslinked PS particles were 153.86 ± 3.65 nm and 163.71 ± 7.74 nm, respectively (Figure 2a,b). Similarly, the cationic surfactant CTAB produced monodisperse PS spherical particles with mean diameters of non-crosslinked and partially crosslinked PS particles of 80.21 ± 2.06 nm and 63.48 ± 3.94 nm, respectively (Figure 2c,d). It should be noted that the final particle size was related to the critical micelle concentration (CMC) of the surfactants used [27]. Generally, low CMC surfactants yield small styrene monomer micelles, resulting in the formation of small polymer particles [28]. The surfactants used in the experiments were SDS and CTAB with CMC values of 8.3 and 0.9, respectively [29,30,31]. SDS has a CMC over about eight times higher than CTAB, resulting in larger styrene monomer droplets and thus larger PS particles.

### 3.2. Partially Etched PS NPs

Chemical etching was used to produce partially etched particles, as illustrated in Figure 1b and Figure 3a. During the growth process of PS particles, the addition of DVB at a designated time $t_{DVB}$ led to the formation of crosslinked PS-co-DVB copolymer chains mainly from the outer layer of the particles. Thus, the crosslinking reaction primarily occurred in the outer layer, leaving the core region of the particles relatively non-crosslinked. In addition, we expected that the delayed DVB addition partially formed weakly crosslinked regions even in the outer layer. When these partially crosslinked PS NPs were dispersed in an organic solvent that could dissolve PS linear chains, only the non- or weakly crosslinked portions of the particles were locally dissolved, resulting in a partially etched morphology.

The particle shape was adjusted by tuning $t_{DVB}$, as shown in Figure 3b,c. When DVB was added immediately after the initiator addition ($t_{DVB}$ = 0 min), the particle shape remained intact after dispersing them in toluene, demonstrating that these PS NPs were completely crosslinked to form PS-co-DVB copolymer. When $t_{DVB}$ increased to 35 min and 70 min, the resulting particles’ surface was partially etched after solvent treatment. As $t_{DVB}$ further increased to 140 min, the non- or weakly crosslinked portions that could be chemically etched increased, resulting in a more significant degree of structural roughness. When DVB was added after the completion of polymerization (i.e., $t_{DVB}=$ 300 min), the crosslinked regions of the PS NPs were minimally formed, resulting in unstable particles in toluene that formed aggregated clumps.

The morphological changes in partially crosslinked PS NPs ($t_{DVB}=$ 70 min) were further evaluated by treating them with different organic solvents. The study used SDS and CTAB surfactants in the polymerization reaction. When examining the PS NPs polymerized in the presence of SDS (Figure 4a–c), the use of THF and chloroform produced a similar morphology compared to toluene. In contrast, the PS NPs polymerized with CTAB were more vulnerable to solvent treatment (Figure 4d–f). When the particles were dispersed in THF, chloroform, or toluene, they formed aggregates and fused together, indicating that an inefficient crosslinking reaction occurred for $t_{DVB}=$ 70 min.

### 3.3. Evaluation of Pickering Emulsion Stability

The stability of oil-in-water Pickering emulsions, which were stabilized with partially etched PS NPs, was compared to those stabilized with intact spherical PS NPs. Note that Pickering emulsions formed with solid particles have a wide range of applications in various fields such as food, cosmetics, pharmaceuticals, oil recovery, and phase-transfer catalysis due to their high stability, biocompatibility, and tunable properties [24,32,33,34,35]. As shown in Figure 5, the emulsions stabilized with the intact spherical PS NPs ($t_{DVB}=$ 0 min sample in Figure 3c) almost disappeared after 2 months, whereas those stabilized with partially etched PS NPs ($t_{DVB}=$ 70 min sample in Figure 3c) remained the same after 2 months, demonstrating their extremely robust and long-lasting stability. This enhanced stability can be attributed to the surface roughness of the partially etched particles. The surface roughness of the partially etched particles increases the surface area exposed to each fluid phase (i.e., oil and water), leading to stronger attachment energy of the particles to the interface [19,20]. This increased attachment property can result in more robust and stable emulsions.

## 4. Conclusions

In this study, we have successfully developed a simple and efficient protocol for fabricating partially etched PS NPs using emulsion polymerization and chemical etching. By adjusting the time of adding the crosslinking agent during the polymerization reaction, we were able to control the degree of crosslinking in the particles. The resulting partially crosslinked PS NPs were chemically etched to selectively dissolve the weakly crosslinked portions, resulting in the formation of partially etched particles with increased surface area. Furthermore, we evaluated the partially etched NPs as solid surfactants to form Pickering emulsions. The resulting emulsions demonstrated significantly improved stability compared to intact spherical NPs, indicating that the partially etched NPs have promising potential as stabilizers for emulsion-based applications. In addition, we believe that the randomly shaped partially etched NPs can be further utilized as model particles that mimic microplastic particles found in nature for cellular toxicity studies. Microplastics have caused serious environmental problems, and their impact on human health is still not fully understood. Therefore, using the partially etched NPs as model particles may provide a valuable tool for investigating the toxicity and environmental impacts of microplastics.

## Figures and Tables

**Figure 1 polymers-15-01684-f001:**
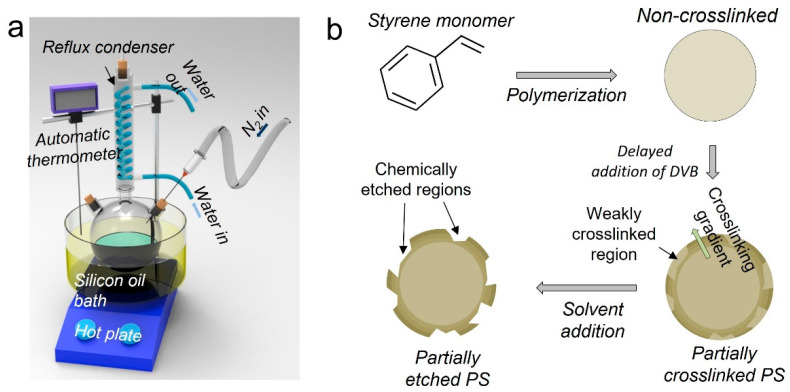
Schematic illustration of fabricating the partially etched PS NPs. (**a**) Experimental setup. (**b**) Proposed mechanism.

**Figure 2 polymers-15-01684-f002:**
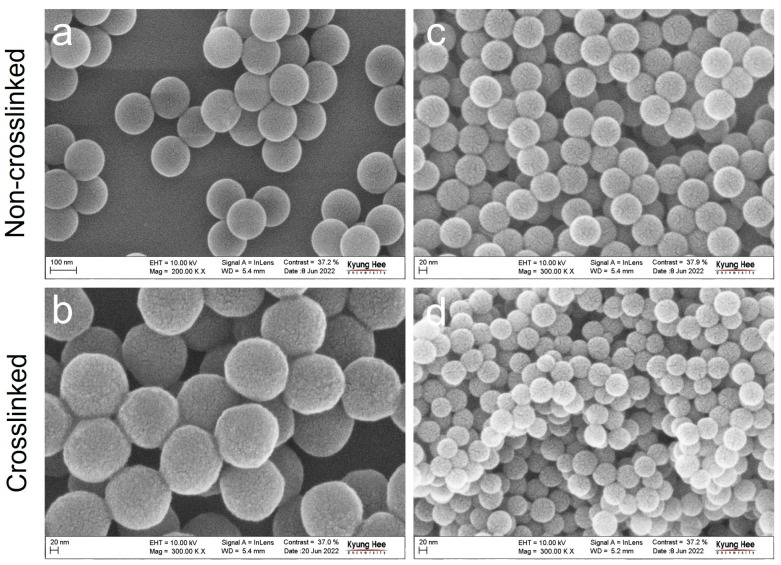
SEM images of PS NPs. Different surfactants were added: SDS (**a**,**b**) and CTAB (**c**,**d**). The top and bottom rows indicate non-crosslinked PS NPs and partially crosslinked PS NPs at $t_{DVD}=$ 70 min, respectively.

**Figure 3 polymers-15-01684-f003:**
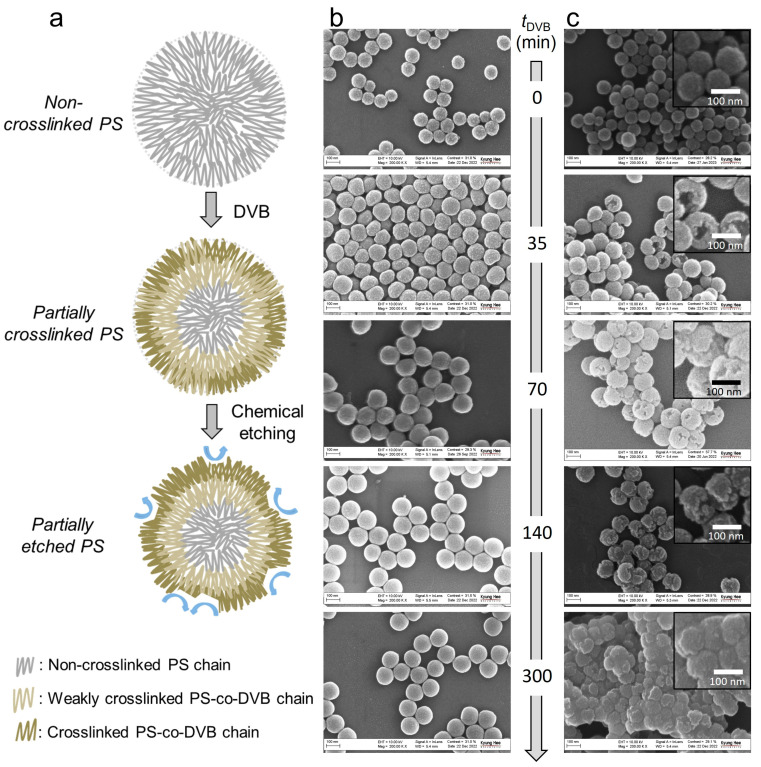
Formation of partially etched PS NPs. (**a**) Schematic for illustrating the prospective mechanism. (**b**,**c**) SEM images showing morphological changes before (**b**) and after (**c**) chemical etching with toluene depending on $t_{DVB}$ (SDS was used). Insets in panel c indicate the corresponding magnified images.

**Figure 4 polymers-15-01684-f004:**
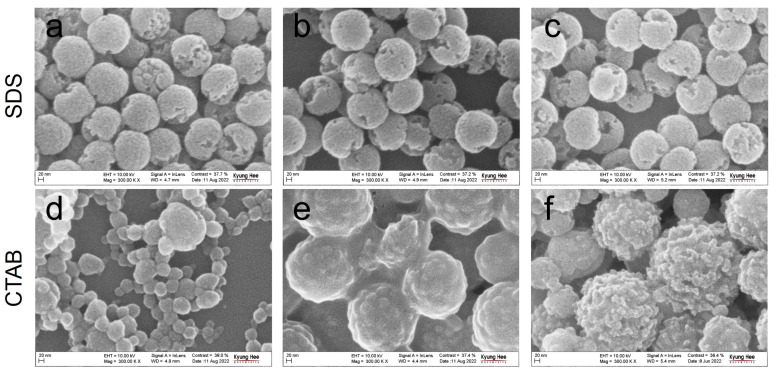
SEM images of partially etched PS NPs after chemically etching them using different solvents: THF (**a**,**d**), chloroform (**b**,**e**), and toluene (**c**,**f**). The top and bottom are the PS NPs polymerized in the presence of SDS and CTAB surfactants, respectively, where $t_{DVB}=$ 70 min for all the cases.

**Figure 5 polymers-15-01684-f005:**
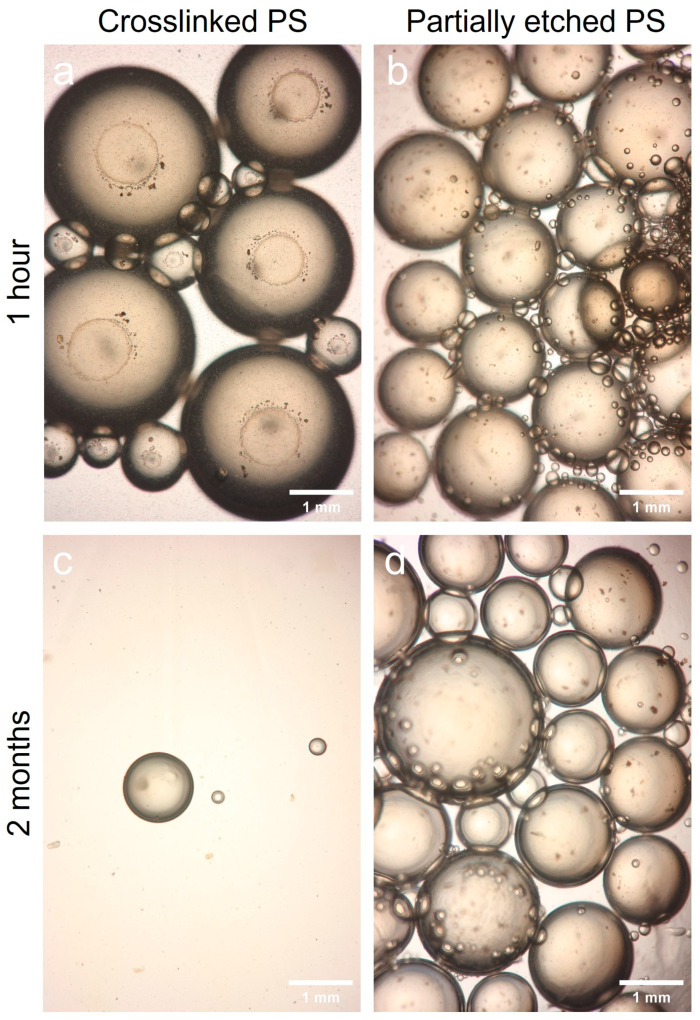
Microscopic images showing the Pickering emulsion stability formed with spherical PS NPs ($t_{DVB}=$ 0 min sample shown in Figure 3c) (**a**,**c**) and partially etched PS NPs ($t_{DVB}=$ 70 min sample shown in Figure 3c) (**b**,**d**).

## Data Availability

All data are included in the manuscript.

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
