# Peer review of "Fabrication of Partially Etched Polystyrene Nanoparticles"

_polymers, 2023, doi:10.3390/polym15071684_

Round 1

Reviewer 1 Report

This work presents a protocol for fabricating partially deformed PS NPs using emulsion polymerization and chemical etching.

(1) Is it possible to give some microcosmic characterization on a single partially crosslined PS, showing the weakly crosslinked region and non-weakly corsslinked region.

(2) I think the word 'deformed'tends to represent the case of morphology change of objects due to the external force (like punch, beat).It is more suitable in the situation of physical change. However, in this paper, the morphology change of the PS is due to the chemical etchng.

(3)Research work which also focus on the polymer based sensor are suggested to be taken into consideration, for example:doi:10.1021/acsnano.2c02609,doi:10.1021/acsnano.7b04158,doi:10.1016/j.matt.2019.11.004;

(3)How about the uniformity between different batches?

Reviewer 2 Report

The manuscript is entitled " Fabrication of partially deformed polystyrene nanoparticles ". Authors have reported the fabrication of partially deformed polystyrene nanoparticles to use them as solid surfactants. However, there are some points that need to be corrected. Therefore, recommended the publication of this paper after minor revision.

Abstract

The first sentence of the abstract section should be impressive and generally cover the content. Then, the content of the study should be mentioned. Please make the necessary corrections.

The abstract should be expanded. First of all, the analysis should be briefly mentioned. Afterward, the future usage areas or the contribution to science should be mentioned with the analysis results.

Introduction

A certain order should be created in the introduction section. How and according to what the specified polymer classes (nonspherical, ellipsoidal, or mesoporous) are grouped should be shared with the reader. First, the polymer classes should be mentioned, and the desired polymer class should be detailed. Then, please make the necessary changes.

The preliminary information given to the reader in the introduction is insufficient. Therefore, the introduction part should be reviewed and detailed according to keywords.

Materials and Methods

The chemical formulas of the materials should be indicated in parentheses. Please make the necessary changes.

The materials used as initiators and crosslinkers should be clearly stated in the sentence, and their abbreviations should be indicated in parentheses.

Please notice that for all commercial material used, please add the information of "manufacture, city, state abbrev if USA, country" for all the software used, please add the information of "version, manufacture, city, state abbrev if USA, country."

Result and Discussion

For the study to be more valuable and remarkable, characterization studies may need to be increased. For example, information can be given about the chemical bonds of the obtained NPs, their thermal resistance, etc.

In addition, necessary tests for cellular toxicity studies mentioned in the conclusion section can be performed and added to the result section.

Reviewer 3 Report

The submitted paper entitled "Fabrication of partially deformed polystyrene nanoparticles" focused on the development of a new method for PS nanoparticle preparation. In my opinion, the presented study is suitable for publication after some minor revision. The subject of study and the presented research concept is novel; the application perspective is also well-defined.

My most crucial comment refers to the introduction section, where the authors did not devote much attention to a detailed literature analysis of the topic.

Besides, it would be useful to explain the mechanism of Pickering emulsion formation and stabilization using the proposed PS-based particles. 
